REGISTERED REPORT

# Registered report: The CD47-signal regulated protein alpha (SIRPa) interaction is a therapeutic target for human solid tumors

Denise Chroscinski[1], Nimet Maherali[2], Erin Griner[3], Reproducibility Project: Cancer Biology*†

[1]Noble Life Sciences, Gaithersburg, Maryland, United States; [2]Harvard Stem Cell Institute, Cambridge, Massachusetts, United States; [3]University of Virginia, Charlottesville, United States

**Abstract** The Reproducibility Project: Cancer Biology seeks to address growing concerns about reproducibility in scientific research by conducting replications of 50 papers in the field of cancer biology published between 2010 and 2012. This Registered report describes the proposed replication plan of key experiments from 'The CD47-signal regulatory protein alpha (SIRPa) interaction is a therapeutic target for human solid tumors' by *Willingham et al., 2012*, published in *PNAS* in 2012. The key experiments being replicated are those reported in Figure 6A–C and Table S4. In these experiments, *Willingham et al., 2012* test the safety and efficacy of anti-CD47 antibody treatment in immune competent mice utilizing a syngeneic model of mammary tumor growth in FVB mice. The Reproducibility Project: Cancer Biology is a collaboration between the Center for Open Science and Science Exchange, and the results of the replications will be published in *eLife*.

*For correspondence: tim@cos.io

Group author details
†Reproducibility Project: Cancer Biology
See page 8

## Introduction

Phagocytosis is an essential process utilized by an organism for pathogen or apoptotic cell clearance (*Poon et al., 2014*). CD47 is a cell surface glycoprotein with a variety of functions including regulation of phagocytosis through binding to the macrophage and dendritic cell specific protein signal regulatory protein alpha (SIRPα) (*Oldenborg, 2013*). Binding of SIRPα to CD47 essentially sends a 'don't eat me' message to macrophages by initiating signaling to inhibit phagocytosis (*Murata et al., 2014*).

Increased expression of CD47 is proposed to be a mechanism through which cancer cells evade immune detection and phagocytosis. CD47 expression is increased in several cancer types including acute myeloid leukemia (AML), acute lymphoblastic leukemia (ALL), non-Hodgkin lymphoma (NHL), primary effusion lymphoma, multiple myeloma, leiomyosarcoma, and bladder cancer, and targeting of CD47 on cancer cells with an anti-CD47 blocking antibody can promote phagocytosis by macrophages in vitro (*Chan et al., 2009*; *Jaiswal et al., 2009*; *Majeti et al., 2009*; *Chao et al., 2010a*; *Edris et al., 2012*). Further, treatment with an anti-CD47 blocking antibody synergized with rituximab treatment to promote phagocytosis in vitro and to eliminate cancer cells in an in vivo xenograft model of non-Hodgkin lymphoma (*Chao et al., 2010b*). This is supported in two syngeneic murine tumor models, in melanoma and squamous cell carcinoma, where irradiation combined with antisense suppression of CD47 delayed tumor growth (*Maxhimer et al., 2009*). *Willingham et al., 2012* further extend these results to demonstrate that CD47 expression increases in a variety of human solid tumor types and that blocking the SIRPα/CD47 interaction with an anti-CD47 antibody can promote phagocytosis of solid tumor cells in vitro and reduce growth of solid tumors in vivo. While it is not clear if SIRPα signaling is

involved in the antitumor activity of an anti-CD47 antibody, these results indicate that anti-CD47 antibody therapy may be an effective treatment for a variety of solid tumor types (*Soto-Pantoja et al., 2012*).

In Figures 6B, 6C, and Table S4, the safety and efficacy of anti-CD47 antibody treatment are tested in immune competent mice using a syngeneic breast cancer model. MT1A2 mouse mammary cancer cells were implanted in the mammary fat pads of FVB mice and IgG control or anti-CD47 antibody treatment commenced upon detection of palpable tumors. Tumor growth was measured by gross weight and analyzed by immunohistochemistry. *Willingham et al., 2012* showed that anti-CD47 antibody treatment reduced tumor growth and increased lymphocytic infiltration to the tumor site without unacceptable toxicity, thus demonstrating that anti-CD47 therapy is effective in reducing solid tumor growth in immune competent hosts. This key experiment demonstrates that CD47 is a therapeutic target for solid tumors and follows similar reports from the same laboratory that also demonstrated that anti-CD47 antibody treatment reduced growth of primary human cancer xenografts of several hematopoietic cancers and of solid leiomyosarcoma tumors (*Jaiswal et al., 2009*; *Majeti et al., 2009*; *Chao et al., 2010a*; *Edris et al., 2012*). Subsequent reports extended these results to multiple myeloma and primary effusion lymphoma models (*Kim et al., 2012*; *Goto et al., 2014*). These experiments will be replicated in Protocol 1.

## Materials and methods

### Protocol 1: Engraftment of mouse breast cancer cells and treatment with targeted antibodies

This experiment tests the safety and efficacy of anti-CD47 antibody treatment in immune competent mice using a syngeneic model of mammary cancer. This experiment replicates figures 6B, 6C, and table S4 of the original article, which assess tumor growth by weight of the tumor 30 days after implantation, lymphocytic infiltration by immunohistochemistry, and toxicity by blood analysis.

### Sampling

1. Experiment has 2 cohorts:

    A. MT1A2 allograft treated with IgG isotype control.
    B. MT1A2 allograft treated with anti-CD47 clone MIAP410.

2. Experiment will use seven mice per treatment group.

    A. To account for unexpected deaths, seven mice will be used per group to ensure at least five will survive for a minimum power of 80%.
    B. A separate, untreated cohort of three mice will be used to gather baseline readings for the blood analysis.

        I. See 'Power calculations' section for details.

### Materials and reagents

All known differences are indicated by an asterisk, with the originally used item listed in the comments section.

| Reagent | Type | Manufacturer | Catalog # | Comments |
|---|---|---|---|---|
| MT1A2 | Cell line | Original lab | n/a | From original lab |
| Dulbecco's Modified Eagle's Medium, high glucose with HEPES modification | Cell culture | Sigma–Aldrich | D6171 | Included during communication with original authors. Original lab used Gibco catalog # 12430-054 |
| L-glutamine | Cell culture | Sigma–Aldrich | G7513 | |
| Fetal bovine serum | Cell culture | Sigma–Aldrich | F0392 | |
| Penicillin/Streptomycin (100×) | Cell culture | Sigma–Aldrich | P433 | |

*Table 1. Continued on next page*

*Table 1. Continued*

| Reagent | Type | Manufacturer | Catalog # | Comments |
| --- | --- | --- | --- | --- |
| Trypsin-EDTA | Cell culture | Sigma–Aldrich | T3924 | |
| Hank's balanced salt solution | Cell culture | Sigma–Aldrich | H6648 | |
| T75 flask | Labware | Sigma–Aldrich | Z707503 | |
| 50-ml tubes | Labware | Sigma–Aldrich | CLS430290 | |
| 1 M Hepes in normal saline | Buffer | Biowhittaker | 17-737E | Included during communication with original authors |
| BSA, IgG free | Chemical | Jackson Immunoresearch | 001-000-161 | Included during communication with original authors. |
| Kolliphor P188 | Chemical | Sigma–Aldrich | K4894 | Included during communication with original authors. Original lab used Pluronic F-68 which has been discontinued |
| Leibovitz L15 media, no phenol red | Cell culture | Life Technologies | 21083027 | Included during communication with original authors |
| Matrigel Matrix High Concentration* | Cell culture | Corning | 354248 | Original from Becton Dickinson |
| 6–8 week old female FVB mice | Animal model | Charles River | Strain Code: 207 | Original from Jackson Labs |
| 27½G needle | Labware | Sigma–Aldrich | Z192384 | Included during communication with original authors |
| 1-ml syringe | Labware | Sigma–Aldrich | Z192090 | |
| 30½G needle | Labware | Sigma–Aldrich | Z192341 | Included during communication with original authors. Original lab used 31 gauge |
| 1–5% isoflurane | Chemical | Specific brand information will be left up to the discretion of the replicating lab and recorded later | | |
| Anti-CD47 clone MIAP410 | Antibody | Original lab | n/a | From original lab |
| Mouse IgG isotype control | Antibody | Innovative Research | IR-MS-GF | |
| PBS | Buffer | Sigma–Aldrich | D8537 | |
| Hematology analyzer* | Instrument | Idexx Laboratories | ProCyte Dx | Original lab used a Heska HemaTrue |
| Neutral buffered formalin | Buffer | Specific brand information will be left up to the discretion of the replicating lab and recorded later | | |
| Ethanol | Chemical | | | |
| Xylene | Chemical | | | |
| Paraffin | Chemical | | | |
| Carazzi's Hematoxylin | Stain | | | |
| Eosin | Stain | | | |
| Permount | Chemical | | | |

## Procedure

### Notes

A. All cells will be sent for mycoplasma testing and STR profiling.

B. Cells maintained in DMEM supplemented with 4 mM L-glutamine, 10% FBS, 100 U/ml penicillin and 100 µg/ml streptomycin at 37°C in a humidified atmosphere at 5% $CO_2$.

1. Culture MT1A2 cells, count cells, gently spin down, and resuspend in FACS buffer so that 50,000 cells can be injected per mouse.

    A. Prepare cells in FACS buffer at 50,000 cells/75 µl.
    B. FACS buffer (500 ml):

       i. 5 ml 1 M Hepes, pH = 7.4.
       ii. 500 mg BSA, IgG free.
       iii. 680 mg Kolliphor P188.
       iv. 5 ml 100X Pen/strep.
       v. Bring up to 500 ml with Leibovitz L15 media, no phenol red.

2. Add high protein matrigel to obtain 25% vol/vol solution.

    A. Total volume/injection is 100 µl and 50,000 cells.

3. Inject 50,000 cells into the left abdominal mammary fat pad (#4) of 6 to 8-week-old female FVB mice using a 27½ G needle.

    A. Use 1–5% isoflurane at 1–2 l/min to anesthetize the mice.
    B. Total number of mice injected is 14.

4. Check mice until palpable tumors form in at least 12 animals.

    A. Approximately 7–10 days after injection palpable tumors will arise.

5. Randomize mice with palpable tumors to two treatment groups using the following method.

    A. On the day the mice are randomized, measure tumors. Animals with no detectable tumors are excluded from the study.
    B. Animals are ranked according to tumor size, to balance groups for baseline tumor characteristics, and assigned to group 1 or group 2 using an alternating serpentine method. (rank 1 = group 1, rank 2 = group 2, rank 3 = group 1, rank 4 = group 2, etc).

       i. Designation of IgG or CD47 antibody treatment as group 1 or group 2 determined by randomly assigning the two treatments into one block using www.randomization.com. Record seed number.

6. Inject 400 µg of antibody, administered in 100 µl PBS, into mammary fat pad proximal to tumor with an approximate distance of 2 mm to the tumor (do not inject directly into tumor), every other day for 30 days using a 30 G needle.

    C. anti-CD47 clone MIAP410 (mouse IgG1).
    D. mouse IgG isotype control.

7. 5 days after the beginning of antibody injections perform complete blood cell counts to assess treatment toxicity with hematology analyzer.

    A. Collect 0.2 ml blood by retro-orbital bleeding.
    B. Collect 0.2 ml blood from three untreated female FVB mice to gather a baseline reading.

8. After 30 days of antibody treatment, sacrifice mice, dissect, and weigh tumor.
9. Dissected tumors are processed for further analysis.

    A. Immediately, place tissues into 10% neutral buffered formalin overnight at room temperature.
    B. Dehydrate tissues through graded alcohols (50%, 70%, 95% ethanol) and clear xylene.
    C. Infiltrate with paraffin, and then embed tissues in a paraffin block.
    D. Cut paraffin blocks on a microtome with a section thickness of 5 µm.

10. Stain tumor sections with H&E (total: 2 stained sections per tumor).

    A. Deparaffinize sections 2 times in xylene, then rehydrate through graded alcohols (95%, 70%, 50% ethanol) to water.
    B. Stain sections with Carazzi's hematoxylin, then rinse slides in water.

C. Stain sections with eosin.
D. Dehydrate sections through graded alcohols (50%, 70%, 95% ethanol) and then place in xylene.
E. Apply coverslips to slides with Permount and store slides at room temperature.

11. Blindly image stained sections and have images blindly analyzed by a Board Certified Pathologist to analyze lymphocytic infiltration of the tissue sections.

A. Assess absence or presence of tumor infiltrating lymphocytes in at least 10 random fields at high power magnification (×400) and score lymphocytic infiltration using the following system (*Demaria et al., 2001*):

i. 0 = absent.
ii. 1 = minimal.
iii. 2 = moderate.
iv. 3 = brisk.

## Deliverables

1. Data to be collected:

A. Mouse health records: general health, weight, and age at time of transplant, survival.
B. Lab records on time course of tumor formation (noting when tumors become palpable), transplants, and antibody injections (including seed number of randomization).
C. Image of each tumor at harvest.
D. Raw numbers and graph of tumor weight in control and anti-CD47-treated mice (compare to Figure 6B).
E. H&E stained sections from each tumor analyzed (compare to Figure 6C).
F. Pathology report for each section and tumor analyzed.
G. Complete blood cell counts in control and anti-CD47-treated mice.

## Confirmatory analysis plan

This experiment assesses if treatment with an anti-CD47 therapeutic antibody alters breast tumor growth in immune competent hosts and examines the safety of the treatment by looking at blood toxicity. The histological analysis of lymphocytic infiltration of the tumors will be reported for each tumor generated during the study, along with the H&E stained sections. This replication attempt will perform the following statistical analysis.

A. Statistical analysis:

Note: At the time of analysis we will perform the Shapiro–Wilk test and generate a quantile–quantile plot to assess the normality of the data. We will also perform Levene's test to assess homoscedasticity. If the data appear skewed, we will perform an appropriate transformation in order to proceed with the proposed statistical analysis. If this is not possible, we will perform the equivalent non-parametric test.

1. Tumor weight from isotype control treated mice to anti-CD47 clone MIAP410 treated mice.

i. Unpaired two-tailed Welch's *t*-test.

B. Hematological parameters (13 parameters) tested in untreated mice, IgG isotype control treated mice, and anti-CD47 clone MIAP410 treated mice.

1. Two-way ANOVA (3 × 13 design) with the following planned comparisons with the Bonferroni correction:

i. One-way ANOVA of untreated, IgG, and anti-CD47-treated mice for each hematological parameter.
ii. Compare the effect size of the original data to the replication data and use a meta-analytic approach to combine the original and replication effects, which will be presented as a forest plot.

## Known differences from the original study

The replication attempt will not include the anti-CD47 clone MIAP301. The replication attempt will analyze lymphocytic infiltration of the tumors using a scoring system of the H&E stained sections, which was not implemented by the original study, and is included as exploratory analysis. Toxicity will be assessed during the course of the efficacy experiment instead of on a different strain and cohort of mice as the original study, which was determined in BALB/c mice. Additionally, the replication study will analyze blood 5 days after the beginning of antibody treatment, which precedes a total of three antibody treatments at a dose of 400 μg/injection. This is similar to the original study, which analyzed blood 5 days after two successive daily antibody injections of 500 μg/injection. To determine the baseline reading, untreated female FVB mice will be also be analyzed. The Idexx Laboratories ProCyte Dx hematology analyzer will assess the same parameters as the Heska HemaTrue used in the original study. All known differences of materials and reagents are listed in the 'Materials and reagents' section above, indicated by an asterisk, with the originally used item listed in the comments section. All differences have the same capabilities as the original and are not expected to alter the experimental design.

The original study analyzed the data using a Student's *t*-test, however since the original data variance is not homogenous the Welch's *t*-test for comparing the samples will be used instead.

## Provisions for quality control

The cell line used in this experiment will undergo STR profiling to confirm its identity and will be sent for mycoplasma testing to ensure there is no contamination, as well as rodent pathogen screening to ensure there are no detectable pathogens. The anti-CD47 clone MIAP410 will be checked to verify the specificity by ELISA, which is being conducted by the original lab. The retro-orbital bleeds will be performed by a Science Exchange lab with expertise in this technique to minimize stress. Subjective data collection will be performed blinded to the experimental conditions and treatment groups will be assigned in a random manner with the seed number recorded to reproduce the plan. All of the raw data, including the H&E stained sections, will be uploaded to the project page on the OSF (https://osf.io/9pbos) and made publically available.

## Power calculations

### Protocol 1

Summary of original data (provided by original authors).

| Dataset being analyzed | N | Mean | SD |
|---|---|---|---|
| Tumor weight of IgG-treated mice | 5 | 0.1445 | 0.05203 |
| Tumor weight of anti-CD47 clone MIAP410 treated mice | 5 | 0.01224 | 0.002258 |

### Test family

A. 2 tailed Welch's *t* test, difference between two independent means, alpha error = 0.05.

### Power calculations (performed with R software, version 3.1.2) (*R Core Team, 2014*).

| Group 1 | Group 2 | Effect size (Glass' Δ)* | A priori power | Group 1 sample size | Group 2 sample size |
|---|---|---|---|---|---|
| IgG | MIAP410 | 2.541995 | 99.8% | 5 | 5 |

*The IgG control group SD was used as the divisor.

Summary analysis of original data presented in Table S4 (*Willingham et al., 2012*):

### Test family

A. 2-way ANOVA (3 treatments x 13 hematology parameters), Fixed effects, special, main effects, and interactions, alpha error of 0.05.

i. ANOVA analysis performed with R software, version 3.1.2 (*R Core Team, 2014*).
ii. Power calculations performed with G*Power software, version 3.1.7 (*Faul et al., 2007*).

| F(Dfn, Dfd) | Partial $\eta^2$ | Original effect size *f* | Replication total sample size | Detectable effect size *f* |
|---|---|---|---|---|
| F(24,39) = 0.8678 (interaction) | 0.348120 | 0.7307699 | 169* | 0.3895070† |
| F(2,39) = 0.8075 (treatments) | 0.039766 | 0.2035014 | 169* | 0.2415459† |
| F(12,39) = 187.6811 (hematology parameters) | 0.982978 | 7.599178 | 169* | 0.3331365‡ |

*The replication sample size includes 13 parameters from 3 untreated, 5 IgG-treated, and 5 CD47-treated mice.
†The original data did not detect a statistically significant interaction or treatment main effect, making these the detectable effect size with 80.0% power.
‡The original data reported a statistically significant effect for the hematology parameters, which the replication is powered to 99.9% to detect. This is the detectable effect size with 80% power.

## Test family

A. ANOVA, Fixed effects, omnibus, one-way, Bonferroni's correction alpha error of 0.05/13 = 0.00385.

i. Power calculations performed with G*Power software, version 3.1.7 (*Faul et al., 2007*).

| Hematology parameter | F(Dfn, Dfd) | Partial $\eta^2$ | Original effect size *f* | Replication total sample size | Detectable effect size *f* |
|---|---|---|---|---|---|
| WBC | F(2,3) = 0.4975 | 0.249071 | 0.57592 | 13* | 1.5234072 |
| Lym | F(2,3) = 0.3297 | 0.180209 | 0.468853 | 13* | 1.5234072 |
| Mono | F(2,3) = 0.9781 | 0.394709 | 0.8075258 | 13* | 1.5234072 |
| Gran | F(2,3) = 1.0706 | 0.416476 | 0.8448228 | 13* | 1.5234072 |
| HCT | F(2,3) = 3.7673 | 0.715222 | 1.584774 | 13* | 1.5234072 |
| MCV | F(2,3) = 58.2710 | 0.974904 | 6.232735 | 13* | 1.5234072 |
| RDWa | F(2,3) = 96.1000 | 0.984631 | 8.004127 | 13* | 1.5234072 |
| HGB | F(2,3) = 2.0036 | 0.571864 | 1.155728 | 13* | 1.5234072 |
| MCHC | F(2,3) = 83.1450 | 0.982279 | 7.445148 | 13* | 1.5234072 |
| RBC | F(2,3) = 2.9797 | 0.665153 | 1.409411 | 13* | 1.5234072 |
| MCH | F(2,3) = 1.3714 | 0.477612 | 0.956183 | 13* | 1.5234072 |
| PLT | F(2,3) = 0.8536 | 0.362682 | 0.7543709 | 13* | 1.5234072 |
| MPV | F(2,3) = 1.9231 | 0.561798 | 1.132278 | 13* | 1.5234072 |

*The replication sample size includes three untreated, five IgG-treated, and five CD47-treated mice.

## Acknowledgements

The Reproducibility Project: Cancer Biology core team would like to thank the original authors, in particular Stephen Willingham and Jens-Peter Volkmer, for generously sharing critical information as well as reagents to ensure the fidelity and quality of this replication attempt, as well as Frank Graham and McMaster University for facilitating the transfer of MT1A2 cells. We thank Courtney Soderberg at the Center for Open Science for assistance with statistical analyses. We would also like to thank the following companies for generously donating reagents to the Reproducibility Project: Cancer Biology; American Type Culture Collection (ATCC), BioLegend, Cell Signaling Technology, Charles River Laboratories, Corning Incorporated, DDC Medical, EMD Millipore, Harlan Laboratories, LI-COR Biosciences, Mirus Bio, Novus Biologicals, Sigma–Aldrich, and System Biosciences (SBI).

# Additional information

### Group author details

**Reproducibility Project: Cancer Biology**

Elizabeth Iorns: Science Exchange, Palo Alto, California; William Gunn: Mendeley, London, United Kingdom; Fraser Tan: Science Exchange, Palo Alto, California; Joelle Lomax: Science Exchange, Palo Alto, California; Timothy Errington: Center for Open Science, Charlottesville, Virginia

### Competing interests

DC: This is a Science Exchange Associated lab. RP:CB: EI, FT and JL are employed by and hold shares in Science Exchange Inc. The other authors declare that no competing interests exist.

### Funding

| Funder | Author |
| --- | --- |
| Laura and John Arnold Foundation | Reproducibility Project: Cancer Biology |

The funders had no role in study design, data collection and interpretation, or the decision to submit the work for publication.

### Author contributions

DC, NM, EG, Drafting or revising the article; RP:CB, Conception and design, Drafting or revising the article

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
