## [Decision Letter]

Thank you for sending your work entitled “Registered report: The CD47-signal
regulated protein alpha interaction is a therapeutic target for human solid
tumors” for consideration at *eLife*. Your article has been
favorably evaluated by Tadatsugu Taniguchi (Senior editor), a Reviewing editor, and four
reviewers, one of whom is a biostatistician.

The Reviewing editor and the reviewers discussed their comments before we reached this
decision, and the Reviewing editor has assembled the following comments to help you
prepare a revised submission. There was considerable discussion about the proposed
replication. One of the reviewers asked for studies with human patient xenograft tumors
to be included for a comprehensive and representative reproduction of the original
study, but the other reviewers and the editors favor strict reproduction. The key
experiment from the prior publication that suggests therapeutic potential for an
anti-CD47-based strategy is the syngeneic experiment in an immune-competent host, where
the potential toxicity of the antibody against the host could be assessed. This
experiment seems to be the most critical to reproduce as it most closely mimics a
therapeutic scenario. Therefore we support the original proposed design, with comments
on the proposal and revision requests following.

Overview:

Chroscinski et al. propose to replicate the experiments from Figure 6B, 6C, and Table S4
of Willingham et al, to verify the safety and efficacy of anti-CD47 antibody treatment
in immune competent mice utilizing a syngeneic breast cancer model, and to assess the
effects on tumor growth and lymphocytic infiltration. The tumor model consists of MT1A2
tumor cells injected into the mammary fat pad of FVB mice. The proposed protocol is
detailed and carefully considered. Tumors will be weighed after 30 days, and embedded in
paraffin and processed for histological analysis of lymphocytic infiltration. Toxicity
will be determined five days after tumor cell injection by complete blood cell
counts.

The authors appropriately discuss the relevant background for the study, and have also
thoroughly considered all parameters of the proposed experiments, for which they have
also consulted with the original authors. Author consultation allowed for the inclusion
of key parameters including multiple details associated with proper media conditions for
the tumor cells.

Notable differences between this proposed work and the previous study include the
exclusion of anti-CD47 clone MIAP301, which is appropriate given the lack of a
statistically significant effect of this clone in the previous study, and toxicity in
this replicate study will be conducted on the same mice for which efficacy is also being
considered, which appears to be preferable to the previous work that used a different
mouse strain (BALB/c). Blood analysis will also be performed with the same dose of
antibody used to determine efficacy, which also seems preferable. The number of mice is
appropriate, as the authors seek to derive data from the same number of animals as in
the original study (n=5 per group final). Finally the tumor cell line will also be
subjected to STR profiling to verify its identity.

Overall the authors propose to replicate what amounts to the key findings of Willingham
et al, in a carefully considered and complete proposal.

Revisions to the proposal and questions to consider:

1) Methods: In the original report two antibodies reactive against CD47, with different
isotypes were used; one isotype control [not specified which) was used as a control. Why
is only one antibody used in replicate here?

2) Methods: In an immunocompetent mouse, tumor can be cleared via CMC, ADCC,
opsonization, phagocytosis, etc, mediated by a number of different effectors. In the
proposed experiment, the test antibody might mediate these mechanisms in addition to
blocking the CD47-SIRPa interaction, with a similar or partial therapeutic result. Using
an isotype control, therefore, does not yield an interpretable result unless the isotype
antibody binds to the same target cell (optimally CD47) with similar affinity, but does
not block the interaction with SIRPa. Such a control antibody was used in some of the
original experiments in vitro, (PNAS, Figure 3A-C), but apparently the antibody later
became “unavailable” and hence was not included in subsequent in vitro or
any animal experiments. Alternatively, use of the same CD47 antibody with a D265A Heavy
Chain mutation to eliminate FcR binding would be a useful contriol. Without such
controls, at the completion of this replicate, a conclusion that the mechanism is via
blockade of the interaction of CD47 with SIRPa, (as in title) is not possible.

3) Methods: Injection of such large and frequent antibody doses, and directly into the
tumor bed, is an unusual administration plan. Use of an IV or IP administration route
would be advised in addition as a control group.

4) How will injections into the orthotropic site, but not the tumor itself be achieved?
It is not clear what Willingham did from reading the paper.

5) Methods: Step 5, How are mice randomized? Please state method.

---

## [Author Response]

*1) Methods: In the original report two antibodies reactive against CD47, with
different isotypes were used; one isotype control [not specified which) was used as a
control. Why is only one antibody used in replicate here*?

We included only the MIAP410 clone because of the larger effect compared to the MIAP301
clone, which the reviewers have also commented on. Also, the authors shared with us the
isotype control used in the original experiment, which was a mouse IgG isotype control
(the same control antibody used in this replication). The MIAP301 clone is a rat IgG
antibody and thus the isotype control antibody was not matched to the MIAP301 clone host
species, but was for the MIAP410 clone.

*2) Methods: In an immunocompetent mouse, tumor can be cleared via CMC, ADCC,
opsonization, phagocytosis, etc, mediated by a number of different effectors. In the
proposed experiment, the test antibody might mediate these mechanisms in addition to
blocking the CD47-SIRPa interaction, with a similar or partial therapeutic result.
Using an isotype control, therefore, does not yield an interpretable result unless
the isotype antibody binds to the same target cell (optimally CD47) with similar
affinity, but does not block the interaction with SIRPa. Such a control antibody was
used in some of the original experiments* in vitro*, (PNAS, Figure
3A-C), but apparently the antibody later became “unavailable” and hence
was not included in subsequent* in vitro *or any animal experiments.
Alternatively, use of the same CD47 antibody with a D265A Heavy Chain mutation to
eliminate FcR binding would be a useful contriol. Without such controls, at the
completion of this replicate, a conclusion that the mechanism is* via
*blockade of the interaction of CD47 with SIRPa, (as in title) is not
possible.*

We agree adding a control antibody to eliminate FcR binding would be an informative
approach to determine if the mechanism is via blockage of the interaction of CD47 with
SIRPa, however, as the reviewers have already described, Willingham and colleagues did
not use this approach. The Reproducibility Project: Cancer Biology aims to perform
direct replications using the same methodology reported in the original paper. We agree
this additional antibody control would be useful, but is beyond the scope of this
project. As such, we will restrict our analysis to the experiments being replicated and
will not include discussion of the other in vitro experiments (Figure 3A-C; [17]) that are not being
replicated in this study.

*3) Methods: Injection of such large and frequent antibody doses, and directly
into the tumor bed, is an unusual administration plan. Use of an IV or IP
administration route would be advised in addition as a control group*.

We agree adding an additional administration route would be of interest to test if the
effect is dependent on the route of antibody delivery, however, as the reviewers have
already described, Willingham and colleagues did not use this approach. The
Reproducibility Project: Cancer Biology aims to perform direct replications using the
same methodology reported in the original paper. We are attempting to identify a balance
of breadth of sampling for general inference with sensible investment of resources on
replication projects to determine to what extent the included experiments are
reproducible. Thus, while the use of an additional administration route would be of
interest, it would be a conceptual replication.

*4) How will injections into the orthotropic site, but not the tumor itself be
achieved? It is not clear what Willingham did from reading the paper*.

We have contacted the original authors who have provided additional details for how
these injections were performed. The antibody was injected into the mammary fat pad next
to the tumor with an approximate distance of 2 mm to the tumor. We have updated the
manuscript to address this point.

*5) Methods: Step 5, How are mice randomized? Please state method*.

We have updated the manuscript to describe this process.